# *Cinetochilides minimus* sp. nov., a Tiny Benthic Ciliate (Protozoa, Ciliophora) from Brackish Water in Korea

**Ji Hye Moon** [1], **Atef Omar** [2] **and Jae-Ho Jung** [1,*]

1 Department of Biology, Gangneung-Wonju National University, Gangneung 25457, Republic of Korea
2 Industry Academy Cooperation Group, Gangneung-Wonju National University, Gangneung 25457, Republic of Korea
* Correspondence: jhjung@gwnu.ac.kr

**Abstract:** During a field survey of Korean marine and brackish water ciliate diversity, we collected a tiny benthic ciliate (13–18 μm long in vivo) from the opening of a brackish water lagoon (10‰). At low magnification, it resembles members of the genus *Aspidisca* because of the oval body shape and the benthic life style, but is not thigmotactic. Based on the observations of living cells, silver-impregnated specimens (i.e., using protargol, silver carbonate, and wet silver nitrate), SEM images, and the 18S rRNA gene sequences, we confirmed that it is a new member of the genus *Cinetochilides*. The new species, *C. minimus* sp. nov., can be easily distinguished from other congeners mainly by the fragmented somatic kinety 1. In spite of the small size, the new species has more than 200 basal bodies, including those in the oral apparatus. The arrangement of the ciliary pattern is rather confusing because of the polymerized kinetids, the sparse basal bodies, the non-ciliated area on the dorsal side, and the presence of parasomal sacs next to the kinetosomes. In the present study, we provide a detailed morphological description and infer the phylogenetic position of *Cinetochilides minimus* sp. nov. based on 18S rRNA gene sequences.

**Keywords:** Cinetochilididae; *Cinetochilum*; lagoon; Oligohymenophorea; protist; taxonomy

## 1. Introduction

The genus *Cinetochilides* Foissner, 2016, consists of four species (type species: *C. terricola* Foissner, 2016) inhabiting saline environments [1–4]. As inferred from the genus name, its morphology resembles that of *Cinetochilum* Perty, 1849, but they can be distinguished by a single morphological (polymerized kinetids in somatic kineties 1 and 2) and two ontogenetic (two rounds of basal body production along the paroral membrane; protomembranelle 1 is elongated and sigmoid) characteristics [3].

For the genera *Cinetochilides* and *Cinetochilum*, the NCBI GenBank database currently includes 18S rRNA gene sequences only for *Cinetochilides ovalis* (Gong and Song, 2008) Foissner, 2016 (basionym: *Cinetochilum ovale* Gong and Song, 2008) and *Cinetochilum margaritaceum*. The sequence of *Cinetochilides ovalis* shows a sister relationship with the subclass Apostomatia; thus, it is distinct from typical scuticociliates [5]. Poláková et al. [4], who redescribed *Cinetochilum margaritaceum* (Ehrenberg, 1831) Perty, 1849 (type species), using morphological data and 18S rRNA gene sequences, erected the new family Cinetochilididae for *Cinetochilides* as type genus mainly based on the molecular phylogeny. Although the two genera have similar morphologies, they are clustered away from each other in the gene tree (and the non-monophyly is supported by the approximately unbiased test).

During a field survey, we collected a tiny benthic, non-thigmotactic ciliate from the opening of a brackish water lagoon; at low magnification it resembled *Aspidisca* species because of the oval body shape and the benthic life style. Based on the observations of living cells, silver-impregnated specimens, SEM images, and 18S rRNA gene sequences, we confirmed that it is a new member of the genus *Cinetochilides*.

## 2. Materials and Methods

### 2.1. Sample Collection and Identification

*Cinetochilides minimus* sp. nov. was discovered in a coastal water sample (salinity of 10‰) at the opening of the Songjiho Lagoon, Republic of Korea. By gently stirring the sandy sediment, the water sample of about 500 mL was taken in January 2022 and transferred to the laboratory within a few hours. The raw culture was maintained in a Petri dish for two months at room temperature (15–20 °C). Sterilized wheat grains were supplied to enrich bacterium as a food source.

The new species was examined under a stereomicroscope (SZ11; Olympus, Tokyo, Japan) and light microscopes (BX53, IX73; Olympus) using bright field and differential interference contrast (DIC) observations at magnifications of 50–1000×. The protargol impregnation 'procedure A' was conducted using synthesized protargol and an acetone developer [6,7]. The silver carbonate and 'wet' silver nitrate impregnation techniques were conducted using the methods used by Foissner [6]. The general terminology follows that used by Foissner [3].

### 2.2. Scanning Electron Microscopy

Cells isolated from the raw culture were prepared for scanning electron microscopy following the protocol used by Foissner [6]. Briefly, they were fixed for 30 min using a mixture composed of 1 part of 2% aqueous osmium tetroxide and 4 parts of concentrated mercuric chloride. The cells were attached on a coverslip using poly-L-lysine and then the coverslip was dehydrated using a graded ethanol series (from 30% to 100%) and dried using a critical point dryer (EM CPD300; Leica, Vienna, Austria). Subsequently, the coverslip was sputter coated with platinum (EM SCD005; Leica) and observed in a JSM-IT500 (JEOL Ltd., Tokyo, Japan).

### 2.3. DNA Extraction, PCR Amplification, and Sequencing

Five cells of *Cinetochilides minimus* sp. nov. were isolated from the raw culture. Each cell was washed more than five times with culture water filtered using a 0.2 μm syringe filter (Minisart® CA Syringe Filters; Sartorius, Goettingen, Germany) and then transferred to a 1.5 mL tube with a minimum volume of water using a glass micropipette. A RED-Extract-N-Amp Tissue PCR Kit (Sigma, St. Louis, MO, USA) was used to extract genomic DNA. The conditions for the PCR testing were as follows: initial denaturation at 94 °C for 1 min 30 s, followed by 40 cycles of denaturation at 98 °C for 10 s, annealing at 58.5 °C for 30 s, extension at 72 °C for 3 min, and a final extension step at 72 °C for 7 min. The 18S rRNA gene was amplified using two primers (New Euk A and LSU rev4) from Jung and Min [8] and Sonnenberg et al. [9]. The PCR amplicons were purified using an MG PCR Purification Kit (MGmed, Republic of Korea). The sequence fragments determined by the New Euk A primer were identical among the five cells; thus, we completed the direct sequencing using one cell. The sequence fragments obtained via direct sequencing were assembled using Geneious 2019.1.8 [10]. The DNA sequencing was performed using an ABI 3700 sequencer (Applied Biosystems, Foster City, CA, USA) and the following internal primers: 18SF790v2, 18SF1470, and 18SR300 [11,12].

### 2.4. Phylogenetic Analyses

To determine the phylogenetic position of the new species, 18S rRNA gene sequences of 101 oligohymenophorean and three colpodean (as outgroup) ciliates were retrieved from the NCBI database. The sequences were aligned using ClustalW [13] and both ends were manually trimmed in BioEdit 7.0.9.0 [14], which consisted of 1622 positions. A jModelTest 2.1.7 [15] was used to select the best-fit model GTR + I + G under the Akaike information criterion (AIC). The maximum likelihood (ML) tree was constructed using IQ-Tree 1.5.3 [16] with 100,000 ultrafast bootstrap replicates. MrBayes 3.1.2 [17] was used for Bayesian inference (BI) analyses with Markov chain Monte Carlo testing for 1,000,000 generations, with a sampling frequency of every 100 generations, while the first 3000 trees were discarded

as the burn-in (average standard deviation of split frequencies = 0.0091, average potential scale reduction factor for parameter values = 1.002). The phylogenetic trees were visualized using the free software package FigTree v1.4.4. The pairwise distances among taxa were calculated in Mega 10.2.4 [18], using the uncorrected p-distance method.

We considered bootstrap values ≥95 as high, from 71–94 as moderate, from 50–70 as low, and <50 as indicating no support [19]. For Bayesian posterior probabilities, we considered values ≥0.95 as high and values <0.95 as low following Alfaro et al. [20].

## 3. Results

### 3.1. Systematics

Phylum Ciliophora Doflein, 1901
Subphylum Intramacronucleata Lynn, 1996
Class Oligohymenophorea de Puytorac et al., 1974
Subclass Scuticociliatia Small, 1967
Family Cinetochilididae Poláková et al., 2021 incertae sedis in the subclass Scuticociliatia
Genus *Cinetochilides* Foissner, 2016
*Cinetochilides minimus* sp. nov.

ZooBank registration number. Present work: urn:lsid:zoobank.org:pub:2D394A02-BA9F-45FA-A1C4-145F902A4BE5, *Cinetochilides minimus* sp. nov.: urn:lsid:zoobank.org:act:AF15D5A6-3777-4A79-B95D-DC795315F989.

### 3.2. Species Diagnosis

Body size on average 15 × 13 μm in vivo, broadly oval to elliptical with notches on posterior cell margin. One macronuclear nodule with 1 micronucleus. Rod-shaped extrusomes along somatic kineties. Usually 13 somatic kineties; somatic kineties 1 and 2 with polymerized kinetids anteriorly; somatic kinety 1 fragmented; somatic kinety 11 with 7 or 8 dikinetids anteriorly. Three adoral membranelles each composed of 3 rows of ciliated basal bodies; two postoral kineties. Brackish water habitat, benthic but not thigmotactic.

### 3.3. Type Locality

The opening of the brackish water lagoon Songjiho (salinity of 10‰; 38°20′8.9″ N, 128°31′21.0″ E), Goseong-gun, Republic of Korea.

### 3.4. Type Material

The slide containing the holotype (MABIK PR00044239) and three paratype slides (MABIK PR00044240–PR00044242) have been deposited in the National Marine Biodiversity Institute of Korean (MABIK). Five other paratype slides (GUC006029, 6040–6043) have been deposited in the Jung lab (J.-H. Jung) in Gangneung-Wonju National University.

### 3.5. Etymology

The Latin species group name *minimus* (masculine; nominative singular) refers to the small body size.

### 3.6. Description

All the descriptions can be found in Figures 1–8.

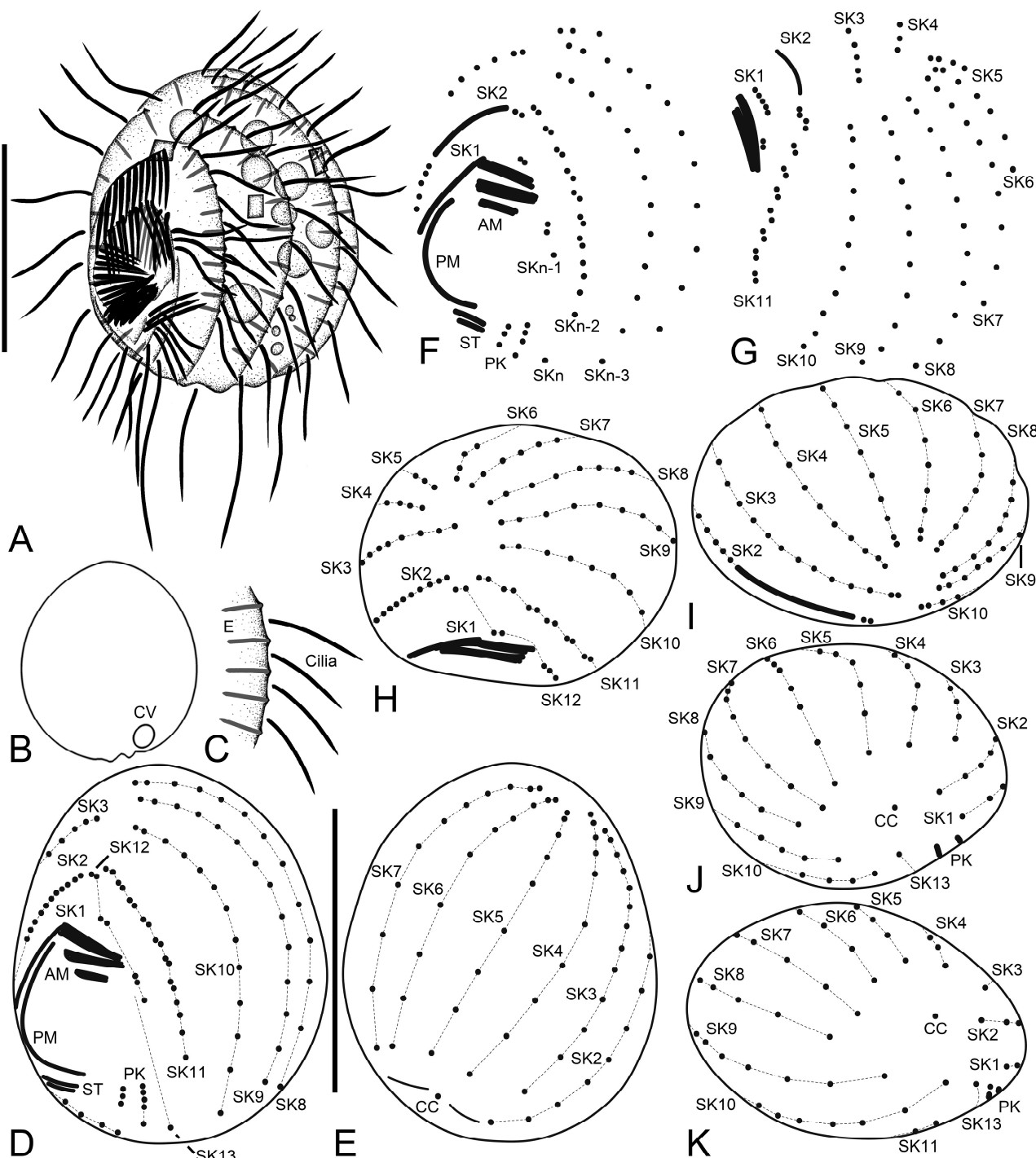

**Figure 1.** *Cinetochilides minimus* sp. nov. from life (**A–C**) and after protargol impregnation (**D–K**). (**A**). A representative specimen showing the body shape, cilia, and extrusomes. (**B**). Dorsal view showing the contractile vacuole. (**C**). Extrusomes. (**D,E**). Ventral (**D**) and dorsal (**E**) view of the holotype. (**F–K**). Ventral (**F**), apical (**G–I**), and posterior polar (**J,K**) views of paratype specimens, showing the basal bodies. Dotted lines connect the basal bodies in each somatic kinety as inferred after wet silver nitrate impregnation, especially for somatic kineties n and n-1. Adoral membranelles, paroral membrane, polymerized kinetids, and scutica are shown in thick black lines because the basal bodies are very close together. AM, adoral membranelles; CC, caudal cilium; CV, contractile vacuole; E, extrusomes; PK, postoral kineties; PM, paroral membrane; SK, somatic kineties; ST, scutica. Scale bars: 10 μm.

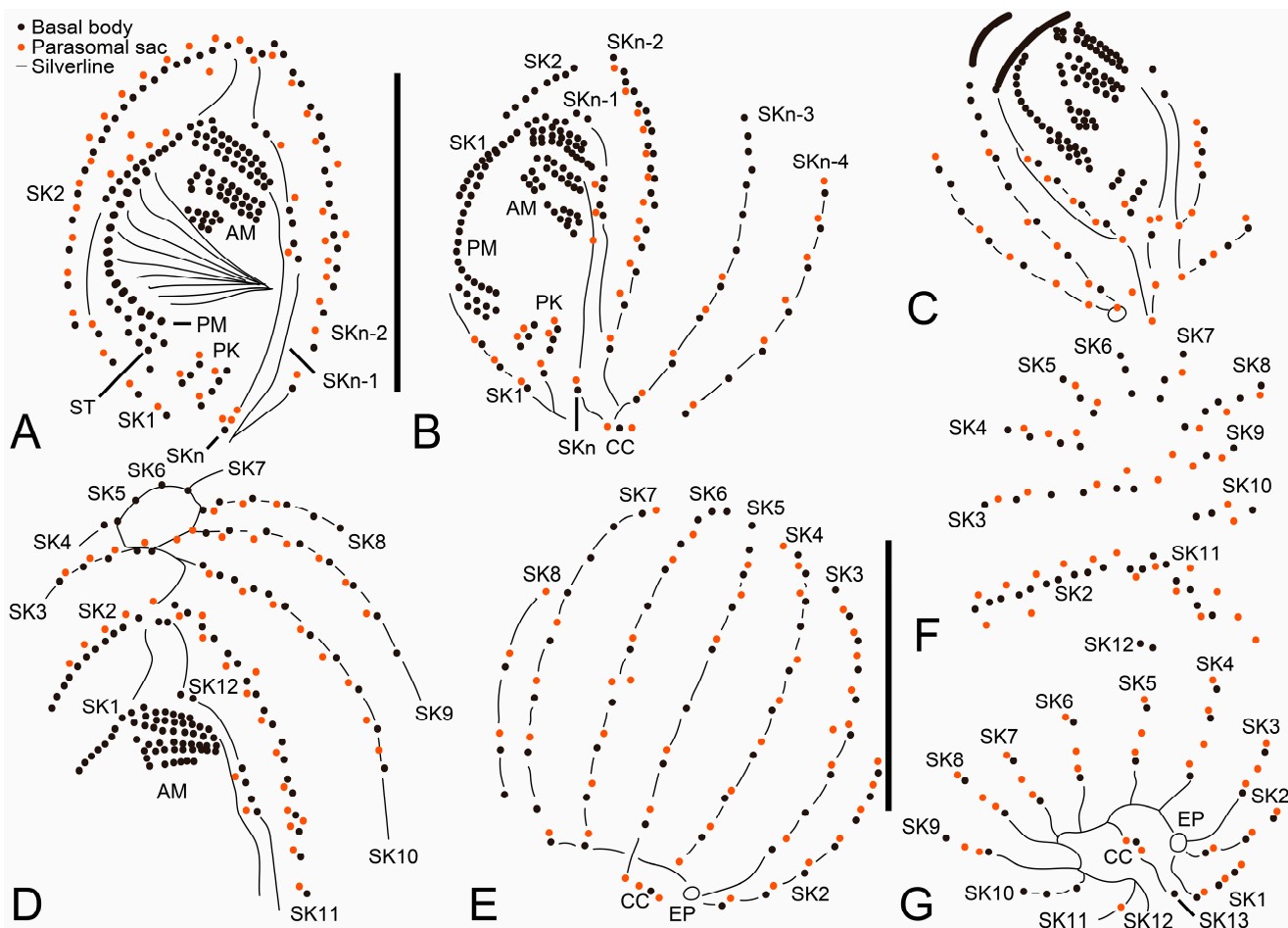

**Figure 2.** *Cinetochilides minimus* sp. nov. after wet silver nitrate impregnation. (**A–G**). Ventral (**A–D**), dorsal (**E**), apical (**F**), and posterior polar (**G**) view of typical specimens. AM, adoral membranelles; CC, caudal cilium; EP, excretory pore; PK, postoral kineties; PM, paroral membrane; SK, somatic kineties; ST, scutica. Scale bars: 10 μm.

Body size 13–18 × 10–15 μm in vivo (n = 29) and 11.4–13.9 × 9.4–11.6 μm (on average 12.6 × 10.6 μm) after protargol impregnation (Figure 1A,B,D,E, Figure 3A–J and Figure 4A–D, Table 1). Body outline broadly oval to elliptical with indistinct notches on posterior cell margin; laterally slightly flattened about 2:1. Nuclear apparatus composed of one elliptical macronuclear nodule and one globular to elliptical micronucleus; macronucleus slightly anterior to mid-body, micronucleus attached to left or lower left margin of macronucleus (Figure 4J,K, and Figure 6A–H). Contractile vacuole subterminal, between caudal cilium and posterior end of somatic kinety 2, with one excretory pore (Figure 1B, Figure 2C,E,G, Figure 3D, Figure 5I, and Figure 7C,E). Extrusomes rod-shaped, about 2.5 × 0.5 μm, usually arranged along somatic kineties; the cortex is slightly crenulate in vivo because of slightly protruding extrusomes (Figure 1A,C and Figure 3I,J). Cortex rigid with longitudinal ridges parallel to somatic and postoral kineties. Cytoplasm colorless, contains a few crystals of various shapes (2–3 μm long) and many food vacuoles (up to 4 μm across). Usually crawling on bottom of Petri dish or upside down at air–water interface; non-thigmotactic, i.e., easily detached from surfaces.

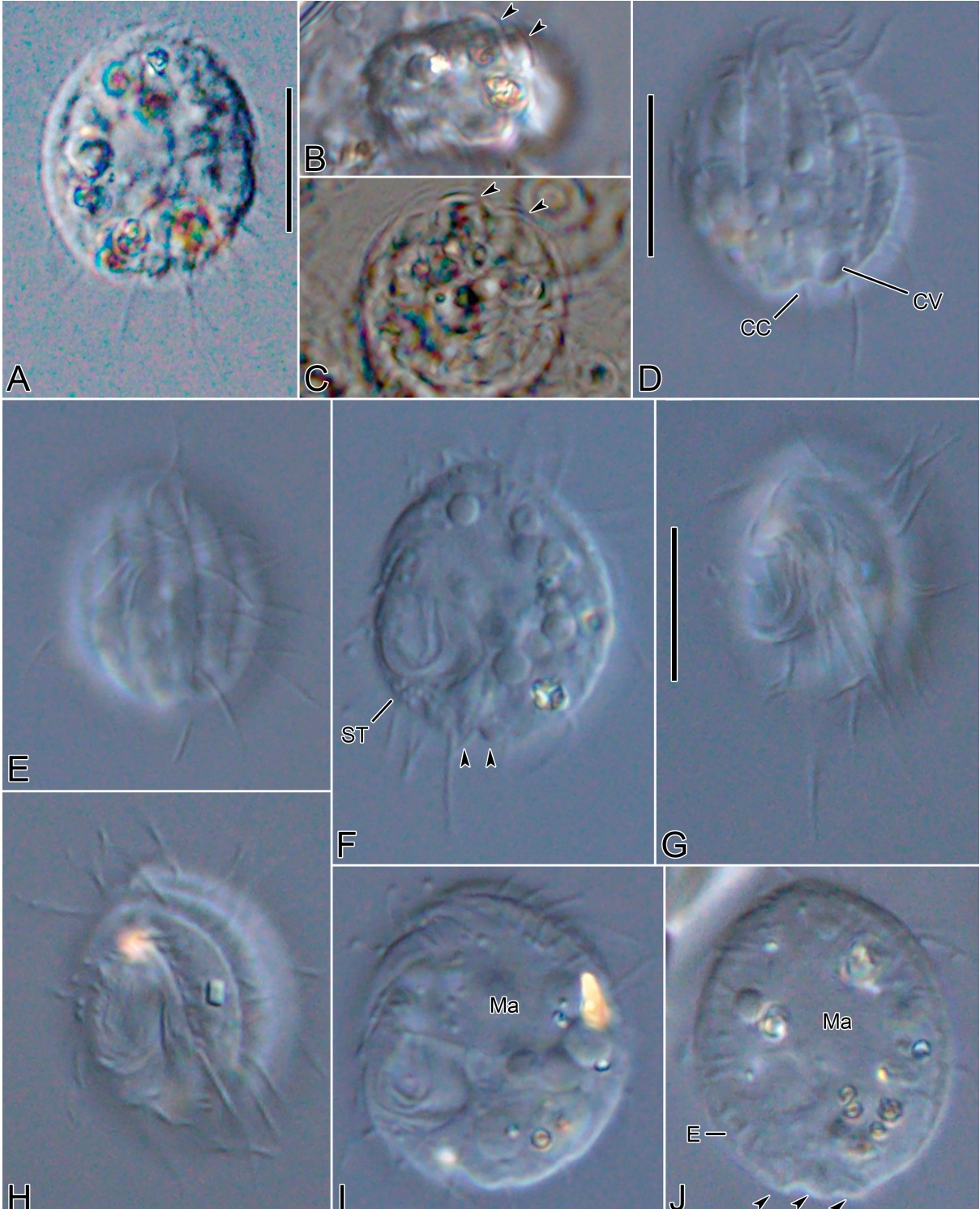

**Figure 3.** *Cinetochilides minimus* sp. nov. from life. (**A**). Ventral view showing the body shape, cilia, and cytoplasmic inclusions. (**B,C**). Apical views showing the dorsal ridges (arrowheads). (**D,E**). Dorsal views showing the contractile vacuole and cilia. Note that posterior ends of dorsal kineties are partially non-ciliated and the caudal cilium has an ordinary length like other somatic cilia. (**F–J**). Ventral views showing indistinct notches (**F,J**, arrowheads), nuclear apparatus (**I,J**), extrusomes (**I,J**), cytoplasmic crystals (**H**), and cilia (**G**). Note that the longest cilia originate from posterior ends of ventral kineties. CC, caudal cilium; CV, contractile vacuole; E, extrusomes; Ma, macronuclear nodules; ST, scutica. Scale bars: 10 μm.

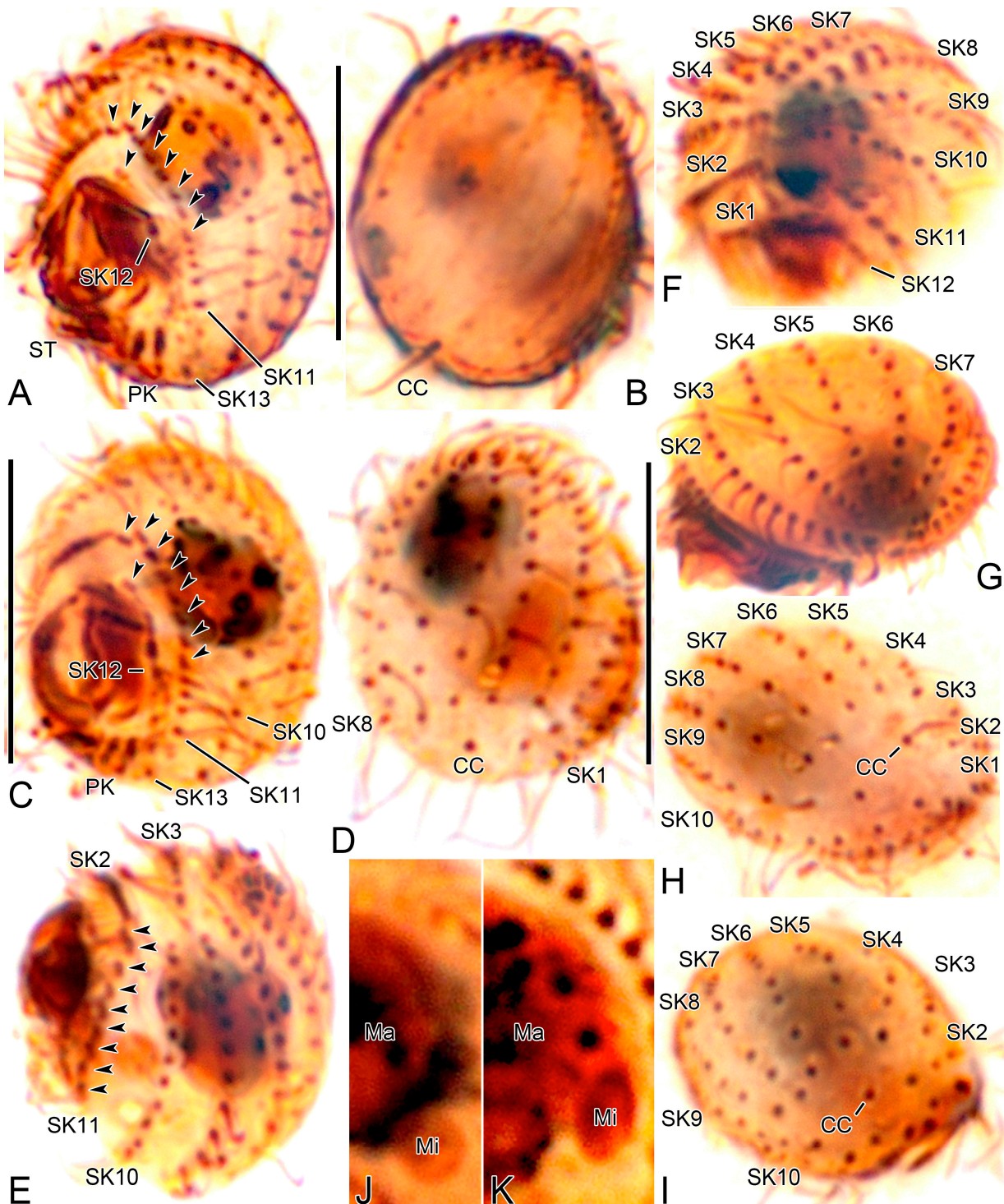

**Figure 4.** *Cinetochilides minimus* sp. nov. after protargol impregnation. (**A**,**B**). Ventral (**A**) and dorsal (**B**) view of the holotype specimen. Arrowheads denote dikinetids in somatic kineties 11 and 12. Note that the caudal cilium has an ordinary length. (**C–I**). Ventral (**C**), dorsal (**D**), left lateral (**E**), apical (**F**,**G**), and posterior polar (**H**,**I**) views, showing the body shape and ciliatures. Arrowheads mark the dikinetids in somatic kineties n-2 and n-1. (**J**,**K**). Nuclear apparatus. CC, caudal cilium; Ma, macronuclear nodules; Mi, micronuclei; PK, postoral kineties; SK, somatic kineties; ST, scutica. Scale bars: 10 μm.

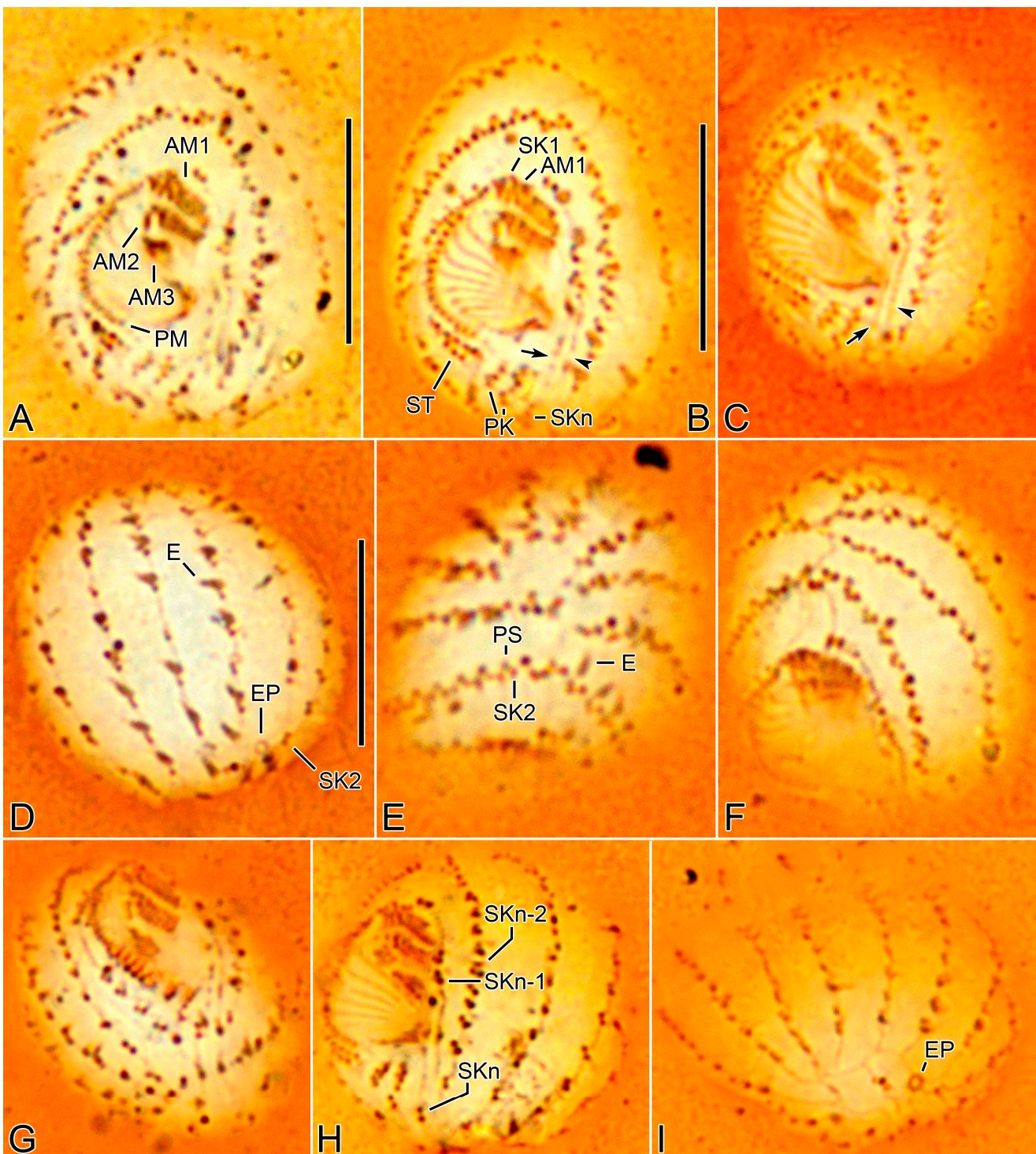

**Figure 5.** *Cinetochilides minimus* sp. nov. after wet silver nitrate impregnation. (**A–C**). Ventral views showing oral and ventral ciliatures. The two kineties SKn and n-1 are arranged along different silverlines (arrows, arrowheads, respectively). (**D**). Dorsal view showing the dorsal kineties, silverlines, and excretory pore. (**E–I**). Apical (**E,F**) and posterior polar (**G–I**) views, showing the anterior and posterior end of somatic kineties, the adoral membranelles, and the parasomal sacs next to kinetids. Note that the first two dikinetids in somatic kinety n-1 are connected by a silverline (**F**). AM, adoral membranelles; E, extrusomes; EP, excretory pore; PM, paroral membrane; PK, postoral kineties; PS, parasomal sacs; SK, somatic kineties; ST, scutica. Scale bars: 10 μm.

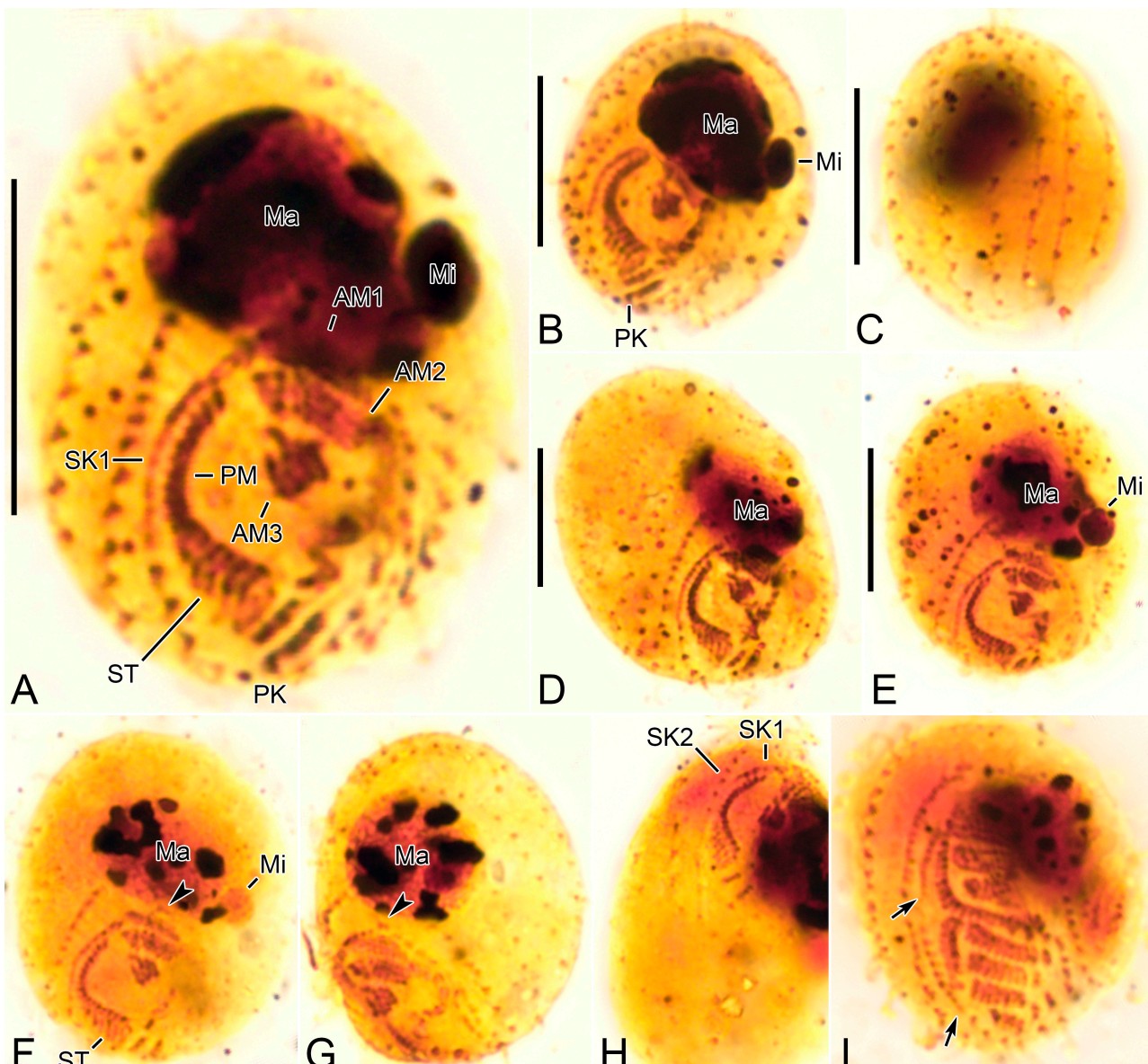

**Figure 6.** *Cinetochilides minimus* sp. nov. after silver carbonate impregnation. (**A–H**). Ventral (**A,B,D–H**) and dorsal (**C**) view showing the ciliatures and nuclear apparatus. The fragmented somatic kinety 1 is connected by kinetodesmata. Note that these specimens were flattened by a coverslip, so that they lost the typical body shape and appeared to be larger than other cells. Arrowheads (**F,G**) denote the dikinetids in somatic kinety n-1. (**I**). Ventral view of mid-divider. The basal bodies of scutica (arrows) are arranged in a row between the paroral membrane and somatic kinety 1. AM, adoral membranelles; Ma, macronuclear nodules; Mi, micronuclei; PM, paroral membrane; PK, postoral kineties; SK, somatic kineties; ST, scutica. Scale bars: 10 μm.

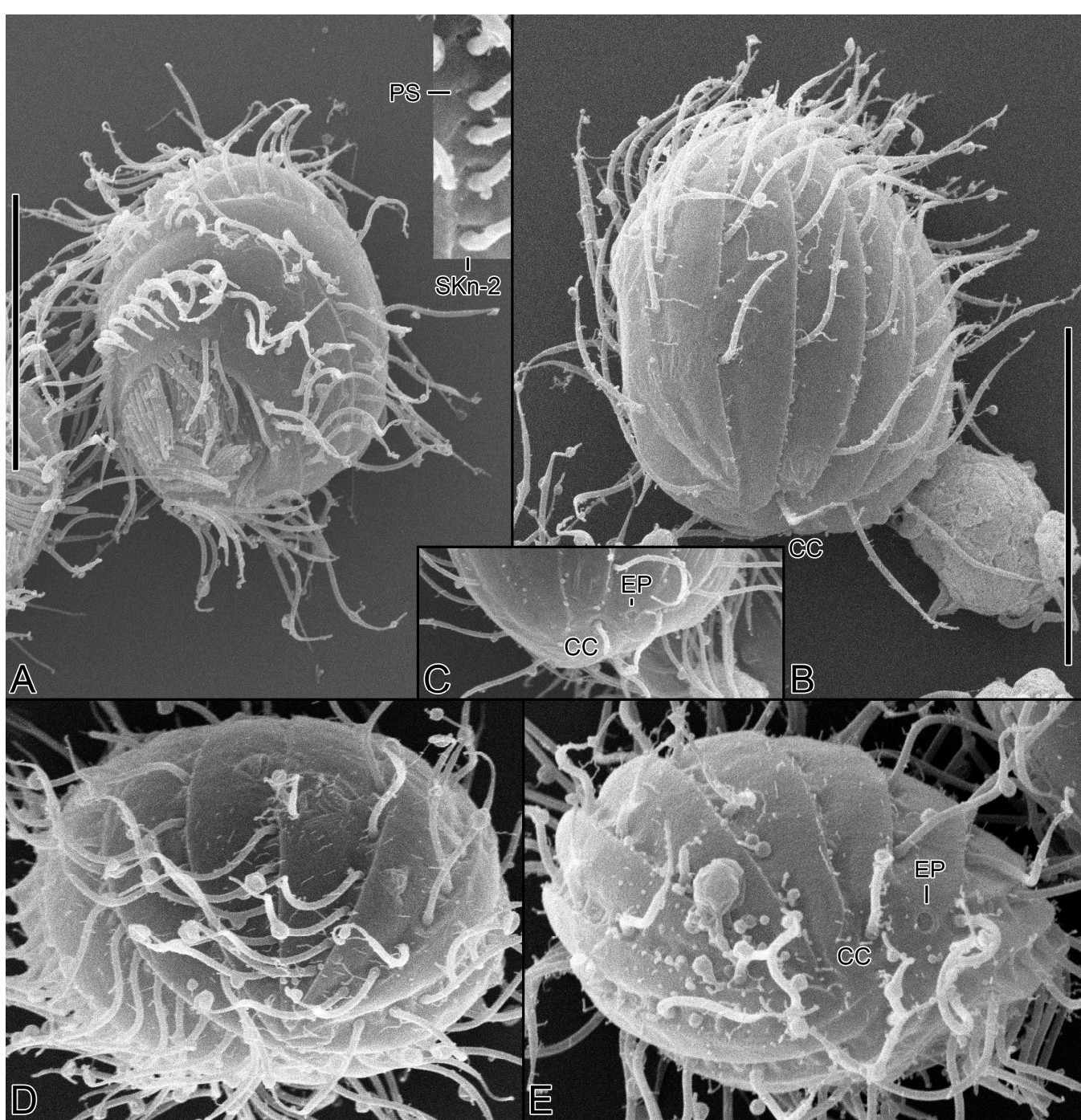

**Figure 7.** *Cinetochilides minimus* sp. nov. in the scanning electron microscope. (**A**). Ventral view of a typical cell showing the body shape, somatic cilia, and the parasomal sacs next to kinetids (inset). (**B,C**). Dorsal views showing the ridges along somatic kineties, the dorsal cilia, and excretory pore of contractile vacuole. The caudal cilium has an ordinary length like other somatic cilia. Note that the posterior ends of dorsal kineties are partially non-ciliated. (**D,E**). Apical (**D**) and posterior polar (**E**) view. CC, caudal cilium; EP, excretory pore; PS, parasomal sacs; SK, somatic kineties. Scale bars: 10 μm.

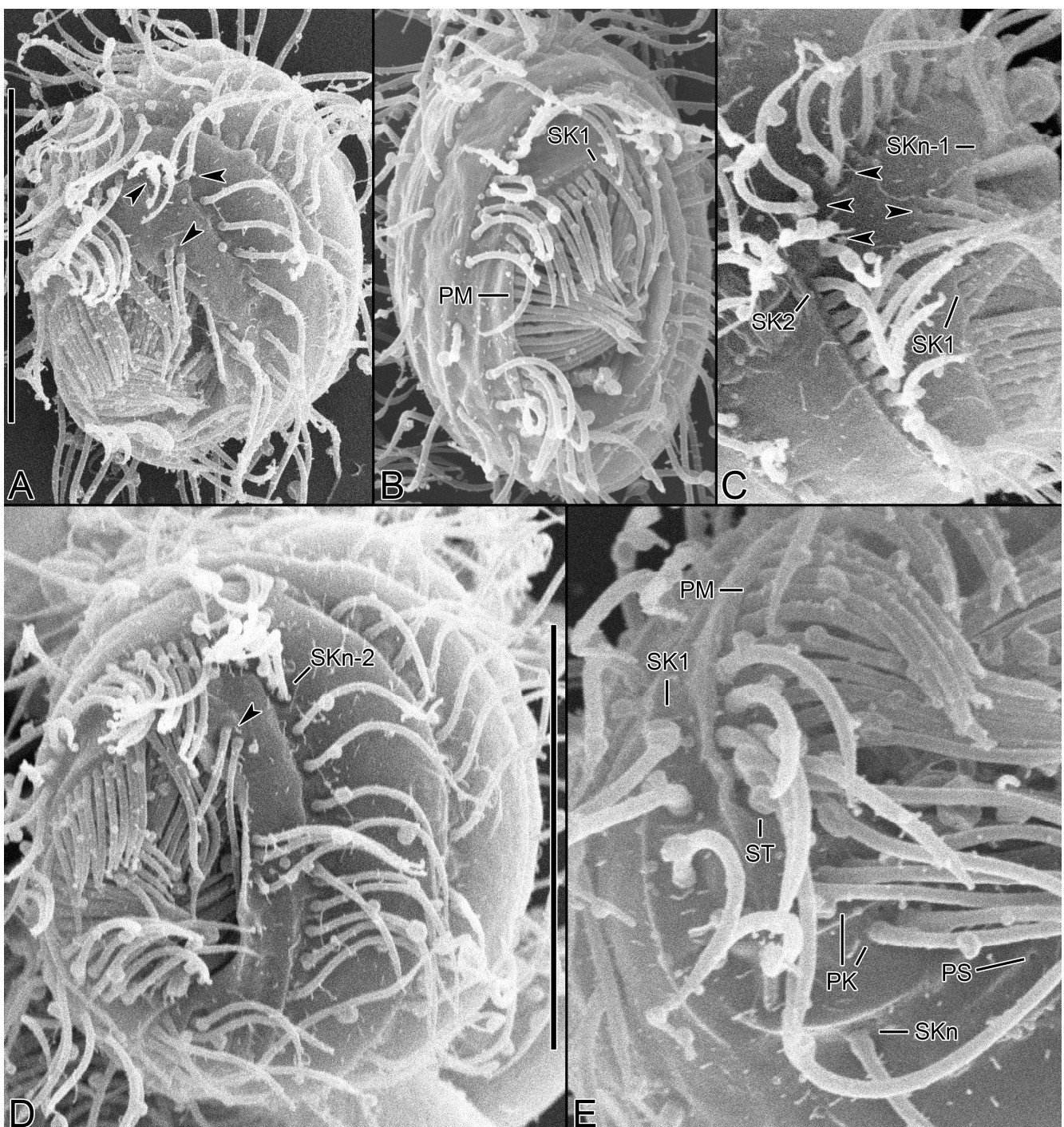

**Figure 8.** *Cinetochilides minimus* sp. nov. in the scanning electron microscope. (**A**–**E**). Ventral views showing the body shape, oral apparatus, and ventral ciliature. Arrowheads denote the dikinetids in somatic kineties n-1 and n-2 and all kinetosomes are ciliated. PK, postoral kineties; PM, paroral membrane; PS, parasomal sacs; SK, somatic kineties; ST, scutica. Scale bars: 10 μm.

**Table 1.** Morphometric data on *Cinetochilides minimus* sp. nov.

| Characteristics [a] | Method | Mean | M | SD | SE | CV | Min | Max | n |
|---|---|---|---|---|---|---|---|---|---|
| Body, length | P | 12.6 | 12.7 | 0.8 | 0.2 | 6.3 | 11.4 | 13.9 | 18 |
| | SN | 16.1 | 16.5 | 1.5 | 0.3 | 9.2 | 12.9 | 18.9 | 19 |
| Body, width | P | 10.6 | 10.8 | 0.6 | 0.2 | 6.1 | 9.4 | 11.6 | 18 |
| | SN | 13.4 | 13.7 | 1.4 | 0.3 | 10.4 | 9.9 | 15.3 | 19 |
| Body length–width, ratio | P | 1.2 | 1.2 | 0.1 | 0.0 | 5.3 | 1.1 | 1.3 | 18 |
| | SN | 1.2 | 1.2 | 0.1 | 0.0 | 9.8 | 1.1 | 1.5 | 19 |
| Anterior body end to proximal end of paroral membrane, distance | P | 9.9 | 10.0 | 0.6 | 0.2 | 6.5 | 8.5 | 10.9 | 14 |
| | SN | 12.1 | 12.5 | 1.5 | 0.4 | 12.3 | 8.8 | 14.3 | 13 |
| Anterior body end to proximal end of paroral membrane, % of body length | P | 79.1 | 79.2 | 4.5 | 1.2 | 5.7 | 69.7 | 87.0 | 14 |
| | SN | 75.1 | 73.8 | 6.5 | 1.8 | 8.6 | 64.5 | 86.2 | 13 |
| Anterior end of somatic kinety 1 to proximal end of paroral membrane, distance | P | 4.9 | 4.9 | 0.3 | 0.1 | 6.6 | 4.2 | 5.5 | 18 |
| | SN | 6.6 | 6.7 | 0.4 | 0.1 | 6.8 | 5.7 | 7.3 | 19 |
| Anterior end of somatic kinety 1 to proximal end of paroral membrane, % of body length | P | 39.4 | 39.1 | 3.1 | 0.7 | 7.8 | 32.6 | 43.6 | 18 |
| | SN | 41.4 | 40.8 | 4.7 | 1.1 | 11.4 | 34.4 | 50.3 | 19 |
| Anterior body end to macronucleus, distance | P | 2.4 | 2.6 | 0.6 | 0.2 | 26.5 | 1.4 | 3.2 | 14 |
| | SN | 3.5 | 3.5 | 0.5 | 0.1 | 14.2 | 2.4 | 4.3 | 13 |
| Anterior body end to adoral membranelle 1, distance | SN | 5.5 | 5.5 | 1.4 | 0.4 | 25.3 | 2.4 | 7.6 | 12 |
| Macronuclear nodules, number | P + SN | 1.0 | 1.0 | 0.0 | 0.0 | 0.0 | 1.0 | 1.0 | 35 |
| Macronuclear nodule, length | P | 5.0 | 5.2 | 0.5 | 0.1 | 10.1 | 3.9 | 5.8 | 18 |
| | SN | 4.7 | 4.6 | 0.5 | 0.1 | 10.5 | 3.9 | 5.8 | 17 |
| Macronuclear nodule, width | P + SN | 4.0 | 4.0 | 0.6 | 0.1 | 14.0 | 3.0 | 5.0 | 18 |
| | SN | 3.2 | 3.1 | 0.5 | 0.1 | 16.4 | 2.4 | 4.0 | 17 |
| Micronuclei, number | P + SN | 1.0 | 1.0 | 0.0 | 0.0 | 0.0 | 1.0 | 1.0 | 6 |
| Micronucleus, length | P | 1.6 | 1.6 | 0.3 | 0.1 | 18.2 | 1.2 | 1.9 | 4 |
| | SN | 1.0 | 1.0 | 0.0 | 0.0 | 0.0 | 1.0 | 1.0 | 2 |
| Micronucleus, width | P | 1.3 | 1.3 | 0.2 | 0.1 | 17.8 | 1.1 | 1.5 | 4 |
| | SN | 0.8 | 0.8 | 0.0 | 0.0 | 0.0 | 0.7 | 0.9 | 2 |
| Somatic kineties, number (without postoral kineties | P + SN | 12.9 | 13.0 | 0.3 | 0.1 | 2.0 | 12.0 | 13.0 | 27 |
| Somatic kinety 1, length of polymerized portion | SN | 4.5 | 4.5 | 0.4 | 0.1 | 9.3 | 3.8 | 5.0 | 18 |
| Somatic kinety 1, number of polymerized kinetids | SN | 12.4 | 13.0 | 1.0 | 0.2 | 8.1 | 11.0 | 14.0 | 17 |
| Somatic kinety 2, length of polymerized portion | SN | 5.9 | 6.0 | 0.6 | 0.1 | 10.3 | 4.9 | 7.1 | 18 |
| Somatic kinety 2, number of polymerized kinetids | SN | 14.6 | 14.0 | 1.6 | 0.4 | 10.8 | 12.0 | 18.0 | 18 |
| Somatic kinety n-2, number of dikinetids | P + SN | 7.1 | 7.0 | 0.3 | 0.1 | 4.5 | 7.0 | 8.0 | 35 |
| Somatic kinety n-2, number of monokinetids | P + SN | 4.0 | 4.0 | 0.5 | 0.1 | 13.6 | 3.0 | 5.0 | 35 |
| Somatic kinety n-1, number of dikinetids | P + SN | 2.0 | 2.0 | 0.0 | 0.0 | 0.0 | 2.0 | 2.0 | 36 |
| Somatic kinety n-1, number of basal bodies below third dikinetid | P + SN | 3.5 | 3.5 | 0.5 | 0.1 | 14.5 | 3.0 | 4.0 | 32 |
| Somatic kinety n, number of basal bodies | P + SN | 1.2 | 1.0 | 0.4 | 0.1 | 32.2 | 1.0 | 2.0 | 37 |
| Postoral kineties, number | P + SN | 2.0 | 2.0 | 0.0 | 0.0 | 0.0 | 2.0 | 2.0 | 37 |
| Adoral membranelle 1, width | SN | 3.2 | 3.2 | 0.3 | 0.1 | 8.1 | 2.5 | 3.7 | 19 |
| Adoral membranelle 2, width | SN | 2.9 | 2.9 | 0.2 | 0.0 | 7.2 | 2.4 | 3.3 | 19 |
| Adoral membranelle 3, width | SN | 1.6 | 1.5 | 0.2 | 0.1 | 14.7 | 1.2 | 1.9 | 19 |
| Paroral membrane, length of chord | SN | 6.5 | 6.4 | 0.5 | 0.1 | 8.0 | 5.8 | 7.9 | 19 |

[a] Data based on protargol-impregnated (P) or wet silver nitrate-impregnated (SN) specimens; all measurements in μm. CV, coefficient of variation (%); M, median; Max, maximum; mean, arithmetic mean; Min, minimum; n, number of specimens examined; SD, standard deviation; SE, standard error of arithmetic mean.

Somatic cilia 5–7 μm long in vivo, including the single caudal cilium at posterior pole of cell; cilia at posterior ends of somatic kineties (SKs) n-3 and n (usually kineties 10 and 13) and left postoral kineties are slightly longer than others, about 8–10 μm. Usually thirteen SKs composed of nine bipolar (SK2–10) and four shortened ones (SK1, 11–13) (Figure 1D–K, Figure 2A–G, Figure 4A–I, Figure 5A–I and Figure 7A–E). SK1 fragmented with a distinct gap between polymerized anterior kinetids (11–14 kinetids) and posterior monokinetids (usually four); anterior three cilia slightly longer than posterior ones, a minute gap between third and fourth kinetids (Figure 2A,B,D, Figure 5A–C,F–H, Figure 6A,D,E,F–H, Figures 7A and 8A–D, Table 1). SK2 composed of polymerized kinetids anteriorly (usually 14 kinetids) and monokinetids posteriorly. SK3–12 begin with a dikinetid(s), but individual kinetids of the dikinetids in SK3–10 are rather loosely arranged. Only posterior basal body ciliated in anterior dikinetids of SK3–10; first two dikinetids in SK11 and first two dikinetids in SK12 are completely ciliated. 'Preoral dikinetids' mentioned by Foissner [3] are very likely homologous to the anterior dikinetids in SK11 and 12. Excretory pore located on the extension of SK2 and 3 (Figure 2E,G and Figure 5D,I). Posterior ends of dorsal kineties (SK3–7) are partially non-ciliated (Figures 3E and 7B,C,E). SK3 is slightly lengthened anteriorly and slightly shortened posteriorly. SK11 (SKn-2) is distinctly shortened posteriorly and usually composed of seven dikinetids anteriorly and four basal bodies posteriorly. SK12 (SKn-1) is distinctly shortened posteriorly and composed of two dikinetids anteriorly and three or four basal bodies posteriorly (probably includes one or two dikinetids); the kinetids are loosely arranged but connected by a silverline (Figure 2A,B,D and Figure 5A–C,F,H).

SK13 (SKn) is composed of one or two kinetids (Table 1); at first glance, the kinetids appear to belong to SK11/12 (SKn-2/n-1), but they are arranged on a different silverline.

Oral apparatus right of midline, oral opening rounded with truncated left end (Figure 1A,D, Figure 3F,H,I, Figure 4A,C, Figure 5A–C,F–H, Figures 7A and 8A,D). Three adoral membranelles, each composed of three rows of basal bodies, decreasing in size posteriorly. Adoral membranelle 1 sometimes with first row of basal bodies shortened at one or both ends compared to second and third rows; note that first three basal bodies of SK1 are ahead of membranelle 1, so that it could be misinterpreted as those of membranelle 1. Adoral membranelle 2 with a cleft at right end. Adoral membranelle 3 roughly quadrate; a minute cleft only observable in silver carbonate preparations (Figure 6A,B,D–G). Paroral membrane C-shaped, sometimes optically intersects anteriorly with the polymerized SK1, composed of dikinetids. About ten oral ribs convergent to pharyngeal fibers (cyrtos-like structure). Scutica at posterior end of paroral membrane, composed of two rows of basal bodies. Invariably two postoral kineties parallel to each other.

Silverline system located along each somatic kinety row. At anterior and posterior ends of cell, silverlines form complete and incomplete circle, respectively, by connecting each somatic kinety with the next one. SKn does not connect with SK1 and SKn-1 posteriorly. Excretory pore of contractile vacuole at posterior ends of kineties 2 and 3 and connects with the silverline of kinety 4 with a transverse line. Caudal cilium kinetid connects with a line from SKn and a silverline from between somatic kineties 5 and 6 (Figure 2A–G and Figure 5A–I). The silverlines are rather weakly developed in the wet silver nitrate preparations, so that the 'dry' silver nitrate impregnation might resolve the issue.

### 3.7. 18S rRNA Gene Phylogeny

The 18S rRNA gene sequence of *Cinetochilides minimus* sp. nov. is 1532 base pairs long with a GC content of 45.4% (GenBank acc. no. OQ164849). The phylogenetic trees using ML and BI analyses show rather similar topologies; thus, only the ML tree was used (Figure 9). The 18S rRNA gene sequence of *C. ovalis* is the only available sequence among the family Cinetochilididae in GenBank. Both sequences clustered together with full support from the ML and BI. They showed an uncorrected pairwise similarity of 97.51%. *Cinetochilum margaritaceum* nested in a clade distant from *C. ovalis + C. minimus* sp. nov. as a sister to the superclade of all other Oligohymenophorea. *Cinetochilum margaritaceum* and *Cinetochilides minimus* sp. nov. showed an uncorrected pairwise similarity of 87.95%.

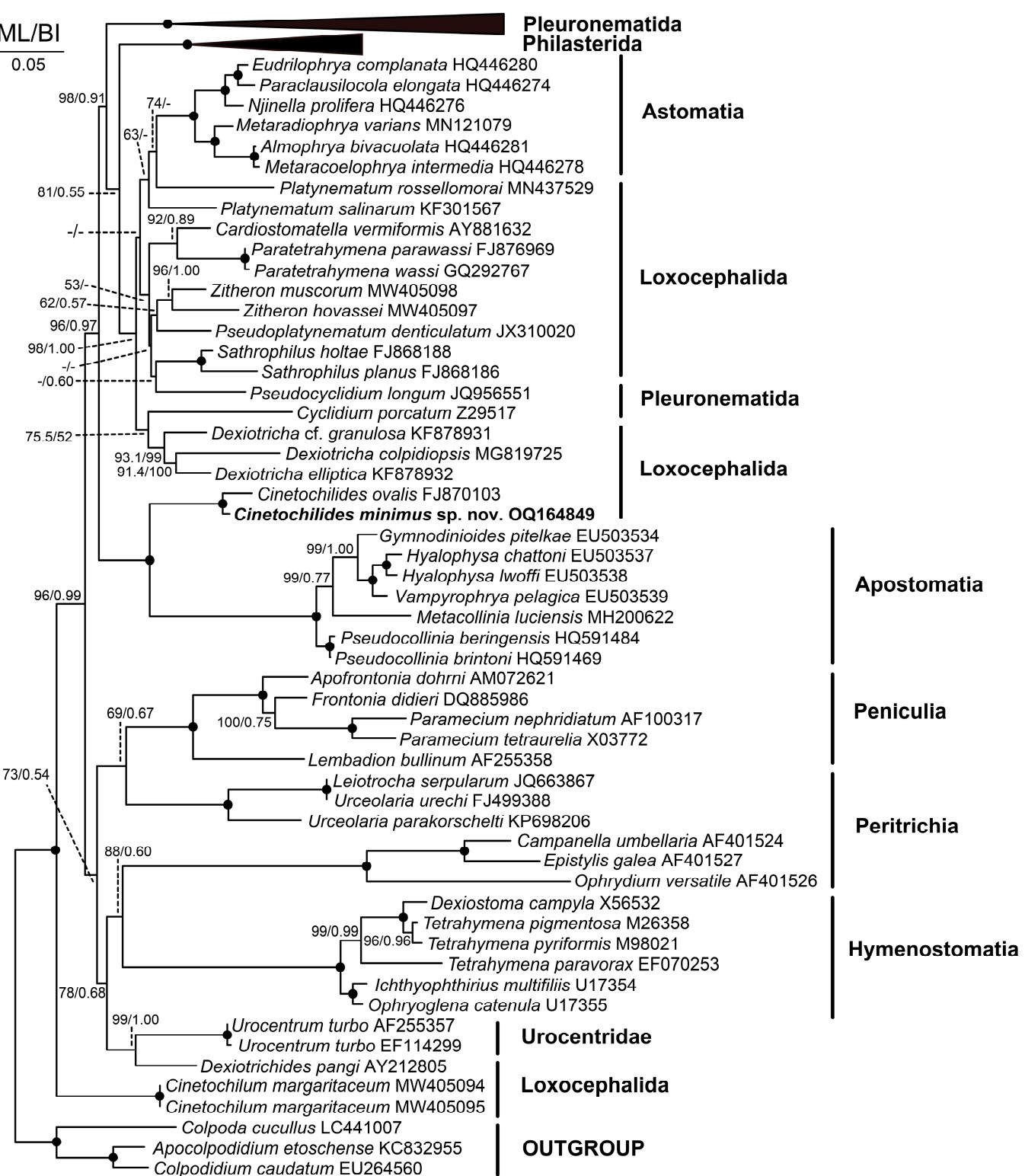

**Figure 9.** Maximum likelihood (ML) tree based on the 18S rRNA gene sequences showing the phylogenetic relationships of *Cinetochilides minimus* sp. nov. The new species is denoted in bold. GenBank accession numbers follow the species names. Numbers at the nodes denote the ML bootstrap values and the posterior probabilities of Bayesian inference (BI), while dots at nodes represent 100 bootstrap values and a 1.00 posterior probabilities. Dashes indicate that the supporting values were less than <50% for ML, <0.5 for BI, or the topologies inferred from BI and ML analyses were incongruent. The scale bar represents five nucleotide substitutions per 100 nucleotides.

## 4. Discussion

### 4.1. Comparison with Related Species

The genus *Cinetochilides* consists of five species including *C. minimus* sp. nov., which inhabit saline soils (*C. australiensis* (Foissner et al., 1994) Foissner, 2016, *C. monomacronucleatus* Foissner, 2016, *C. terricola*) and marine littoral areas (*C. ovalis*, *C. minimus* sp. nov.) [1–3]. They are all small in size (usually less than 30 μm long in vivo) and have similar body shapes, requiring stained specimens for species identification. *Cinetochilides minimus* sp. nov., however, has a fragmented somatic kinety 1 (vs. non-fragmented in other congeners) that easily discriminates the new species from the congeners (Table 2).

**Table 2.** Comparison of *Cinetochilides minimus* sp. nov. with congeners.

| Characteristics | *C. australiensis* (Type Pop.) | *C. monomacronucleatus* (Type Pop.) | *C. ovalis* (Type Pop.) | *C. terricola* (Type sp., Type Pop.) | *C. minimus* sp. nov. |
|---|---|---|---|---|---|
| Body, length in vivo | ~28 μm | 15–25 μm | 20–30 μm | 24–30 μm | 13–18 μm |
| Macronucleus, numbers | 2 | 1 | 1 | 2 | 1 |
| Somatic kineties, number | 11 | 11 or 12 | 12 or 13 | 11 | 12 or 13 |
| Fragmentation of somatic kinety 1 | Absent | Absent | Absent | Absent | Present |
| Scutica, position | Behind paroral membrane | Behind paroral membrane | Behind somatic kinety 1 | Behind paroral membrane | Behind paroral membrane |
| Membranelle 1, structure | Continuous | Continuous | Fragmented | Continuous | Continuous |
| Horizontal silverlines | Present | N/A | N/A | N/A | Absent |
| Habitat (salinity) | Saline soil (20–25‰) | Saline soil (~50‰) | Sandy littoral area (30‰) | Saline soil (~30‰) | Sandy littoral area (10‰) |
| Reference | Pomp and Wilbert [1] | Foissner [3] | Gong and Song [2] | Foissner [3] | Present study |

In addition to the fragmentation of somatic kinety 1, which is the main diagnostic key of *C. minimus* sp. nov., there are other key features supporting the validity of the new species, as follows. *Cinetochilides australiensis* differs from *C. minimus* sp. nov. by the body length (28 μm vs. 13–18 μm in vivo), the number of macronuclear nodules (2 vs. 1), and the horizontal silverlines (present vs. absent) [1]. *Cinetochilides monomacronucleatus* can be distinguished from *C. minimus* sp. nov. by the number of dikinetids in somatic kinety n-2 (2 vs. 7 or 8) and the length of somatic kineties n–n-2 (long vs. short) [3]. Note that Foissner [3] mentioned that *C. monomacronucleatus* and *C. terricola* are morphologically indistinguishable except for the number of macronuclei (1 vs. 2). *Cinetochilides ovalis* can be discriminated from *C. minimus* sp. nov. by the body length (20–30 μm vs. 13–18 μm in vivo), the position of the scutica (behind somatic kinety 1 vs. behind paroral membrane), and the non-fragmented somatic kinety1 (vs. fragmented) [2].

### 4.2. Phylogeny Based on 18S rRNA Gene Sequences

The genus *Cinetochilides* was established by Foissner [3]. It has a similar morphology to the genus *Cinetochilum*, but he separated them by the polymerized kinetids in somatic kineties 1 and 2 and two ontogenetic features (two rounds of basal body production along the paroral membrane; protomembranelle 1 elongated and sigmoid). Next, Poláková et al. [4] established the family Cinetochilididae (incertae sedis in subclass Scuticociliatia) and assigned *Cinetochilides* as the type genus (monotypy) based mainly on the 18S rRNA gene phylogeny. As such, the monophyly of the two genera was rejected by the approximately unbiased test. However, the 18S rRNA gene sequence of the type species (*C. terricola*) is unavailable for measuring the genetic similarity and assessing the congeneric assignment. Poláková et al. [4] split the non-monophyletic group Loxocephalida into 10 lineages (I–X), and Loxocephalida I and III each includes *Cinetochlum* and *Cinetochilides*, respectively.

Of the congeners of *Cinetochilides*, only the 18S rRNA gene sequence of *C. ovalis* is available from GenBank. It clustered with *C. minimus* sp. nov. with full support in the phylogenetic tree (Figure 9). As in previous studies, they show a sister relationship with Apostomatia [4,5]. The apostomes are a monophyletic group primarily adopted to a symbiotic life history [21].

**Author Contributions:** Conceptualization, J.H.M., A.O. and J.-H.J.; data curation, J.H.M. and J.-H.J.; funding acquisition, J.-H.J.; investigation, J.H.M., A.O. and J.-H.J.; methodology, J.H.M. and J.-H.J.; visualization, J.H.M. and J.-H.J.; writing—original draft, J.H.M. and J.-H.J.; writing—review and editing, J.H.M., A.O. and J.-H.J. All authors have read and agreed to the published version of the manuscript.

**Funding:** This research was supported by National Marine Biodiversity Institute of Korea (2022M01100, 2022M00200).

**Data Availability Statement:** The data presented in this study can be found in online repositories. The names of the repository or repositories and accession number(s) can be found in the article.

**Acknowledgments:** We are grateful to the Center for Research Facilities at Gangneung-Wonju National University for their assistance in the analysis of the cell structure (CPD, SEM, etc.).

**Conflicts of Interest:** The authors declare no conflict of interest.

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
