# Peer review of "Cinetochilides minimus sp. nov., a Tiny Benthic Ciliate (Protozoa, Ciliophora) from Brackish Water in Korea"

_diversity, doi:10.3390/d15010076_

Round 1
Reviewer 1 Report
This is a well documented description of a new species of Cinetochilides. Because the systematic status of the Loxocephalida is currently far from resolved this paper is of significant interest. The morphologic aspects of the new species are meticulously documented in the photomicrographs and drawings, all of which are high quality. The attached pdf includes comments and questions that should be addressed and some suggestions for revising the text for clarity. These are merely suggestions. The description could be modified a bit to reflect the somewhat greater variability in some features of the infraciliature as depicted than is suggested in the text. The phylogeny can improved by considering in a bit more detail the strongly supported position of Cinetochilum as sister to all other Oligohymenophora.

Author Response
Dear reviewer 1,
we truly appreciate the comments to improve our MS and all the authors mentioned in the manuscript have participated in the research and have read research content very carefully. According to the comments on the attached pdf, we have revised this MS.
Sincerely yours,
Jae-Ho.
Reviewer 2 Report
The manuscript describes a novel ciliate from the brackish waters in Korea. The identification of the species as new to science seems correct. The methodology used is appropriate and photomicrographs and line diagrams are of good quality. The silver nitrate stained images and SEM photographs provide further details along with morphometric tables in support of the study presented. The English is also good. Thus I have only minor suggestions for this manuscript that may be considered prior to acceptance.
1. Table 1 is not cited in the text.
2. similarly figure 5 is also not cited in the text. Please check for other figs.
3. scale bars could be added to figs 2 and 8.
4. 18. S rDNA could be written as 18S r DNA
5. The GenBank accession no. of the new species could be added.
6. Zoo bank registration number to be added.
Author Response
Dear reviewer 2,
we truly appreciate the comments to improve our MS and all the authors mentioned in the manuscript have participated in the research and have read research content very carefully.
Reviewer's comments:
1. Table 1 is not cited in the text.
A: Revised.
2. similarly figure 5 is also not cited in the text. Please check for other figs.
A: Revised.
3. scale bars could be added to figs 2 and 8.
A: Revised.
4. 18. S rDNA could be written as 18S r DNA
A: Revised. To unify the term, we have used '18S rRNA gene'.
5. The GenBank accession no. of the new species could be added.
A: We have registered our gene sequence to the GenBank. Because it takes days, we will include the number during proofreading process.
6. Zoo bank registration number to be added.
A: Added.
Sincerely yours,
Jae-Ho.
Reviewer 3 Report
A new species of scuticociliates is discovered and described, which is a new contribution to the taxonomy and biodiversity studies of ciliates. The figures and drawings are of high quality, in particular, the good staining and detailed morphometrics of such a small ciliate are commendable. Both the description and comparison are clear enough and the validity of the new species is with no doubt.
I suggest to provide the description of the silver-line of the caudal end. There is a line going through the caudal cilium basal body, where it comes and which somatic kineties it connects is rather stable.
Author Response
Dear reviewer 3,
we truly appreciate the comments to improve our MS and all the authors mentioned in the manuscript have participated in the research and have read research content very carefully.
Reviewer: I suggest to provide the description of the silver-line of the caudal end. There is a line going through the caudal cilium basal body, where it comes and which somatic kineties it connects is rather stable.
A: According to the comment, we have included the description about the silverline.
Sincerely yours,
Jae-Ho.